# A Study on the Mediating Role of Emotional Solidarity between Authenticity Perception Mechanism and Tourism Support Behavior Intentions within Rural Homestay Inn Tourism

**DOI:** 10.3390/bs12090341

**Published:** 2022-09-16

**Authors:** Jie Chen, Chang Liu, Yuqi Si, Rob Law, Mu Zhang

**Affiliations:** 1Shenzhen Tourism College, Jinan University, Shenzhen 518053, China; 2College of Tourism, Fujian Normal University, Fuzhou 350108, China; 3Asia-Pacific Academy of Economics and Management, Department of Integrated Resort and Tourism Management, Faculty of Business Administration, University of Macau, Macau 999078, China

**Keywords:** rural homestay inn, Yunshui Yao, authenticity perception, emotional solidarity, tourism support behavior intentions

## Abstract

Rural homestay inns are an important part of rural tourism, and tourists’ support behavior intentions are important factors affecting whether rural homestay inns can be developed sustainably. The local authentic life experiences and realization of actual communication between the host and tourists are the main influencing factors for tourists to revisit, recommend, or provide support. Although previous studies have confirmed the influence of authenticity perception on tourists’ support behavior intentions from different perspectives, they have not analyzed the influence mechanism between them from the perspective of micro interpersonal emotional attitude. To further understand the impact mechanism between the two, this study introduces the variable of emotional solidarity; constructs a relationship model of authenticity perception, emotional solidarity, and tourists’ support behavior intentions by adopting the theory of reasoned action; and verifies the established hypotheses through empirical analysis. The results show that both existential authenticity and objective authenticity positively influence tourism support behavior intentions, and the effect of objective authenticity on tourism support behavior intentions is greater than that of the presence of authenticity. Empathic understanding, feeling welcome, and emotional intimacy all play mediating roles between intrapersonal authenticity perception and tourism support behavior intentions. Findings also show empathic understanding and feeling welcome play mediating roles in objective authenticity perception and between the perception of objective authenticity and tourism support behavior intentions. Suggestions are also proposed for the development of homestay inn enterprises.

## 1. Introduction

As rural homestay inns are important subjects of rural tourism, the construction path of tourists’ behavior intentions to support these establishments should be understood to achieve a sustainable development of these inns and promote the prosperity of the rural tourism industry. The realization of local authentic life experiences and the genuine interactions between the host and tourists are crucial factors that influence tourists to revisit, recommend, or provide support [1]. Tourists expect to experience and feel the authentic emotions and life scenes of their hosts and the distinctive cultural characteristics of the local area during their stay in rural homestay inns [2,3]. Therefore, the development of authenticity perception (AP) in rural lodging warrants further examination. MacCannell argued that “cultural production and cultural experience in the modernization process are intertwined in all aspects of social life, and as a result, individuals in society develop a strong interest in and curiosity about the authentic lives of others” [4], and the search for authenticity is one of the main tourism motivations of modern tourists [5]. Although an academic consensus on the concept of authenticity is yet to be reached, authenticity in relation to homestay inns has received some research attention [2,6,7,8,9,10]. In addition to meeting the basic needs of tourists in terms of accommodation and catering, rural homestay inns bring tourists the opportunity to experience the unpretentious life of their owners, engage in spontaneous emotional exchanges, and discover their selves. Tourists can further interact and connect with the hosts emotionally under the effect of authenticity, which has been constantly strengthened [11]. As an important concept for explaining the emotional connection among people, emotional solidarity (ES) refers to the identification and recognition of individuals within one another through the pursuit of common values. This identification not only continuously strengthens relationships but also makes them more intimate, thereby promoting a positive perception toward the interactive communication experiences of individuals [12]. Tourists’ support for homestay inns can be regarded as their context-influenced behavior in a particular setting and can be linked through behavior intentions. Previous studies have confirmed the influence of AP on tourism support behavior intentions (TSBI) from different perspectives [13,14,15], but only few have examined the influence of AP on TSBI in rural homestay inns [6,7,16] and the ES between the host and tourist, which is an important outcome of AP and an important driver of positive behavior.

Academics generally believe that the formation of TSBI is a process in which cognition leads to emotion and then affects behavior. Emotional attitudes have a mediating effect between AP and TSBI. Place attachment is often used as an emotional intermediary to discuss tourists’ attitudes toward the whole tourism destination [15,17], and a micro interpersonal perspective toward the relationship between tourists and local hosts remains lacking. It is of both theoretical and practical significance to explore the tourists’ emotional attitudes toward homestay inn owners, an important component of rural homestay inns, in the relationship between AP and TSBI, in order to improve the service quality of rural homestay inn hosts.

The theory of rational action (TRA) originated from social psychology and has been widely used to study and predict the impact of people’s attitudes on behavior. This theory holds that an individual’s behavior will be affected by subjective norms and behavior attitudes. In 1967, Fishbeinhe et al. established the rudiment of the theoretical model of rational behavior in their exploration of attitude. In 1975, Fishbeinhe and Ajzen further perfected this model and finally established the theoretical model of TRA, which posits that an individual’s attitude will be triggered after being stimulated by their external environment. In addition, such behavior can be predicted on the basis of the individual’s pre-existing attitude and TSBI, which follows the transmission logic of “cognition–attitude–behavior [18].” TRA provides a theoretical basis for exploring the transmission mechanism of TSBI from the perspective of AP, which makes the establishment of this model highly scientific and rigorous. Tourists may have “specific” psychological feelings toward the host of their homestay under the experience of authentic elements, which indirectly affect their support behavior intentions. This study refers to the 3D scale of emotional intimacy, empathic understanding, and feeling welcome proposed by Woosnam [19] and uses each of these dimensions as a mediating variable between AP and TSBI to establish a transmission logic of “AP → ES → TSBI.”

In summary, this paper empirically examines whether ES mediates the relationship between AP and TSBI and identifies the specific pathways of the influence mechanism of TSBI. From the tourists’ authenticity perspective, this paper adopts the quantitative research methods of questionnaire survey and structural equation model, introduces ES as an intermediary variable, and discusses the structural dimensions of AP and its impact on ES and TSBI so as to provide theoretical support for further clarifying the connotation of AP, strengthening ES between host and tourist, and improving TSBI. At the same time, this study provides a thinking path for promoting the sustainable development of rural areas.

## 2. Literature Review and Relationship Hypothesis

### 2.1. Authenticity Perception and Tourism Support Behavior Intentions

Academic research on TSBI in the tourism sector has largely focused on revisit intention and willingness to recommend [20]. Revisit intention refers to the possibility of tourists choosing a new destination again for the future and is viewed as the tendency shown by tourists when evaluating their desired destination. Meanwhile, recommendation intention refers to the judgment of tourists on whether they are willing to recommend a destination to others. In the field of tourist behavior, the willingness to revisit and recommend is referred to as TSBI. Therefore, this paper uses “revisiting” and “recommending” as general terms for TSBI.

The concept of authenticity was first applied in heritage studies [21], such as on historical sites [22,23,24], ancient villages [25,26], and folklore [27,28], but as these studies matured and deepened, the concept of authenticity was extended to other fields, including rural homestay inns [29].

Wang Ning divided authenticity into three categories, namely, constructivist or symbolist authenticity, objective authenticity, and existential authenticity [29]. On the basis of the descriptions in the previous section, this paper argues that the AP of rural homestay inn tourists is mainly concerned with both objective authenticity perception (OAP) and existential authenticity perception (EAP). OAP refers to “how people perceive themselves in relation to an object” [3] and includes both the tourist’s desire to visit and experience historical sites and the establishment of authentic knowledge of arts, crafts, and objects [30,31]. The OAP of tourists in rural homestay inns involves all aspects of their perception of the physical environment in the process of accommodation, such as hardware facilities, sanitation, design and decoration of rooms, and catering quality of rural homestay inns.

EAP involves the object freedom of activities or experiences [29,30,32] and covers two parts, namely, the feelings between and within individuals [29]. Among them, the interpersonal part is related to natural emotions, such as feeling the sincere and enthusiastic attitudes of the host or feeling as warm as home. Meanwhile, the inner feeling of the individual is related to the emotion of self-creation [30], such as experiencing the authentic life of the homestay inn host or triggering the thoughts on lifestyle and life attitude.

In other words, in addition to the accommodation functions of a standard hotel, rural homestay inns offer a highly authentic and contextualized experience to tourists. Specifically, the personal feelings that arise when a tourist engages in a non-standardized, host-involving, and warm service that they perceive as the result of an authentic encounter cannot be replicated in traditional hotel settings [33]. This AP is most likely a strong driver of positive behavior after tourism, such as revisiting, recommending it to others, and providing support. The impact of AP on TSBI has been confirmed by previous studies [6,7,8]. The following hypothesis is then proposed:

**H1.** *Authenticity perceptions (dimensions) positively influence tourism support behavior intention*.

### 2.2. Mediating Role of Emotional Solidarity

ES may arise through interaction when people share common or similar beliefs and behaviors [12]. ES refers to a sense of mutual identity among group members due to having the same or similar common value system, which leads to closer and more stable relationships [34]. The concept of emotional solidarity was introduced in tourism research by Woosnam, who built on Durkheim’s theoretical framework of emotional solidarity and argued that destination residents and tourists in the same tourism space can experience positive emotional solidarity through shared beliefs, common behaviors, and interactions.

Woosnam developed the emotional solidarity scale (ESS) [35], which consists of three dimensions, namely, feeling welcome (FW), emotional intimacy (EI), and empathic understanding (EU). FW refers to destination residents taking pride in their local area and the contribution of tourists to the local economy. EI refers to destination residents perceiving a close relationship and building friendship with tourists. EU refers to destination residents feeling a sense of commonality with and closeness to tourists, thereby showing them their understanding and support. ES avoids exploring the relationship between tourists and homestay inn hosts from the perspective of social exchange and instead investigates the construction of host–tourist relationships from the perspective of their emotional connection [35]. ES occurs after the tourist develops their AP, that is, after the tourist perceives a highly authentic interpersonal emotional connection with the attentive and warm hospitality of the homestay inn host as reinforced by the AP.

Rich findings on the influence of ES on TSBI have also been obtained. Previous studies have accordingly used ES as an influencing factor to measure resident or tourist satisfaction and TSBI. For example, Ribeiro et al. found that tourist welcome and empathic understanding directly influence TSBI after a survey of tourists in the Cape Verde Islands [36]. Liang et al. found that ES can indirectly influence TSBI through the mediating role of satisfaction by using heritage tourists as study objects [37]. The following hypotheses are then proposed:

**H2.** *Authenticity perception* (*dimensions*) *positively influences emotional solidarity* (*dimensions*).

**H3.** *Emotional solidarity* (*dimensions*) *positively influences intention to tourism support behavior intention*.

**H4.** *Emotional solidarity* (*dimensions*) *mediates between authenticity perception* (*dimensions*) *and tourism support behavior intention*.

On the basis of the relevant literature and the above hypotheses, the influence mechanism of tourism support behavior intention is modeled, as shown in Figure 1.

## 3. Methodology

This section describes the methods used in this article. The measurement questionnaire was developed and revised before carrying out field research and collecting primary data.

### 3.1. Questionnaire and Measurement Scale Design

The questionnaire was designed in three stages. At the first stage, the questionnaire was designed on the basis of the relevant literature and was divided into seven sections, including an opening greeting, experience-related questions, an authenticity perception scale, an emotional solidarity scale, a tourism support behavior intention scale, and personal information.
(1)The authenticity perception scale, drawing upon the studies of Lalicic and Weismayer [6] and Y. Jiao and H. Xu [1], uses the 2 dimensions of OAP and EAP, and contains a total of 10 items.(2)The measure of emotional solidarity scale, drawing upon the ESS of Woosnam [19], uses the three dimensions of EI, EU, and FW, and includes a total of 11 items.(3)The measure of tourism support behavior intentions scale, which is based on the TSBI scale of Lalicic and Weismayer [6], includes the 3 dimensions of willingness to recommend, willingness to revisit, and provide support, and has a total of 4 items.


At the second stage, to design a questionnaire that is in line with the actual situation of Yunshui Yao, this study analyzed the tourist reviews of 10 rural homestay inns with high ratings (4.5 or above, 5-point scale) in Yunshui Yao, Fujian, on 12 January 2021 through the crawler technology available on Ctrip, an online travel agency. A total of 1152 reviews were reviewed. After filtering out the obviously invalid reviews, which are ratings below 4 or with fewer than 30 words, a total of 504 valid reviews were obtained, and the questionnaire items were revised for the second time through keyword frequency and semantic network analyses. 

At the third stage, after preparing the first draft of the questionnaire, relevant experts were invited to examine the applicability and accuracy of the questionnaire items to which the variables belonged. According to their comments on the overall content and semantics of the scale, those questionnaire items that were semantically repetitive and prone to ambiguity were revised, and the questionnaire design was finalized.

### 3.2. Data Collection

Located in Nanjing County, Zhangzhou City, Fujian Province, China, the Yunshui Yao Ancient Town is an ancient village with a long history. In recent years, with the rapid development of tourism and the changing needs of tourists, many traditional houses in ancient towns have gradually been transformed into homestay inns with local characteristics. The Yunshui Yao homestay inn community is large in scale with dozens of homestay inns, most of which are run by the homeowners themselves, who incorporate many of their own ideas and living concepts into the design and operation of their inns, hence highlighting the authenticity of their establishments. This study selected the Yunshuirang homestay inn as a typical representative of rural homestay inns to test the research hypotheses. The questionnaire items were rated on a five-point Likert scale, ranging from “strongly disagree” to “strongly agree,” being based on the personal feelings of the respondents.

The questionnaire was pretested online on 18 January 2021. A total of 150 questionnaires were sent out, of which 138 valid questionnaires were returned, thereby yielding a valid return rate of 92%. The pretest data were then subject to item and reliability analyses. The item analysis was performed to test the discriminatory power and reliability of the scale items. On the basis of the item analysis results, the questions “I was impressed by the overall construction style of the homestay inn” and “My stay at the homestay inn helped me to relax and release stress” were deleted from the objective authentic dimension. The overall reliability of the questionnaire was 0.945, and the reliability coefficients of each construct ranged from 0.838 to 0.910, thereby indicating the good reliability and trustworthiness of the questionnaire [38]. The formal survey started on 13 February 2021, and the questionnaires were collected through a combination of online and offline methods. The online questionnaires were mainly distributed to netizens who had stayed at the Yunshui Yao homestay inns through various social media channels, whereas the offline questionnaires were distributed onsite in the Yunshui Yao ancient town. As of 15 March 2021, a total of 320 authentic questionnaires were collected. After excluding those questionnaires that were completed in less than 60 s and were obviously invalid, a total of 298 valid questionnaires were retained in this study, thereby yielding an effective rate of 93.13%. The collected data were then analyzed using SPSS 24.0 and Amos 24.0. 

## 4. Results

### 4.1. Demographic Analysis

The sample had a roughly equal proportion of male (40.9%) and female (59.1%) respondents. In terms of age, most of the respondents (215, or 72.2% of the sample) were aged between 19 and 39 years. In terms of educational attainment, undergraduate and tertiary education were the most common (62.1%), followed by high school or junior secondary education (21.5%). In terms of occupational composition, private sector employees accounted for 59.7%, followed by students at 25.5%. In terms of income structure (converted RMB into USD), the largest proportion of respondents (36.91%) earned between USD 723 and USD 1157 per month, followed by those earning more than USD 1446 (20.8%), earning between USD 1157 and USD 1446, and earning between USD 434 and USD 723 (15.4%). In terms of accommodation experience, most of the respondents stayed in homestay inns for one (54.4%) or two nights (37.6%).

### 4.2. Reliability and Validity Analysis

Principal components analysis was performed to conduct exploratory factor analysis of three variables, namely, AP, ES, and TSBI. The results are shown in Table 1. First, the data for AP were rotated several times, and the first two factors all had eigenvalues greater than 1 with a cumulative variance contribution rate of 66.373%. The two principal components were eventually identified, which were labelled as “OAP” and “EAP” according to the results of the literature combing. Second, the data line of ES was rotated several times, and the first three factors all had eigenvalues of greater than 1 and a cumulative variance contribution of 77.989%. These factors were labelled as “EU”, “FW”, and “EI” according to results of the literature combing. Third, an exploratory factor analysis was conducted on the TSBI data, and only one factor had an eigenvalue of greater than 1 with a cumulative variance contribution of 76.043%. The commonality and factor loadings of each questionnaire item were greater than 0.4, thereby meeting the criteria [39]. The reliability of each dimension ranged from 0.863 to 0.910 (all greater than 0.7), and the CITE values all ranged from 0.539 to 0.883 (all greater than 0.5), thereby confirming the strong reliability of the questionnaire items.

Confirmatory factor analysis was performed using maximum likelihood estimation to separately measure the convergent and discriminant validity of the scale. As shown in Table 2, the standardized loading coefficients for each questionnaire item factor were above 0.5, thereby indicating the good fit of the six-factor model (χ^2^/df = 2.544, RMSEA = 0.072 < 0.08, CFI = 0.920 > 0.9, TLI = 0.907 > 0.9, IFI = 0.920 > 0.9). The AVE values ranged between 0.553 and 0.717, which exceeded the recommended value of 0.5, whereas the CR values ranged between 0.859 and 0.907, which exceeded the recommended value of 0.7 [40], thereby confirming the excellent convergent validity of the model. The results of the discriminant validity test are shown in Table 3, where the correlation coefficients between the variables were smaller than the square root of the AVE of each variable itself, thereby indicating that the measurement model has good discriminant validity. In sum, the model had a good overall fit.

### 4.3. Hypothesis Test

#### 4.3.1. Path Analysis

The model was analyzed using AMOS 24.0 and showed a generally good fit to the data (χ^2^/df = 2.698, RMSEA = 0.076, CFI = 0.911, TLI = 0.898, IFI = 0.911). The results in Table 4 show that both EAP and OAP had a significant positive effect on EU and FW, EAP had a significant positive effect on EI, and OAP did not have the same significant positive effect. The two dimensions of AP and the three dimensions of ES all had a significant positive effect on the TSBI. Therefore, H1 and H3 were supported, whereas H2 was partially supported. The hypothesis test results are shown in Figure 2.

#### 4.3.2. Mediating Effect Analysis

Baron and Kenny argued that mediation should meet three conditions. First, the independent variables should have a significant effect on the dependent variables. Second, the independent variable should have a significant impact on the mediating variable. Third, the mediating variable should have a significant effect on the dependent variable [41]. According to the path analysis results, the path coefficient from OAP to EI is not significant. Therefore, the mediating effect of verifying its path no longer warrants consideration. This study used process plug-in, Bayesian estimation, and the bias-corrected percentile bootstrap method to test the significance of the mediation effect when the confidence interval does not include 0 at a 95% confidence interval and after 5000 iterations [42]. Table 5 shows five intermediary paths: EAP → EU → TSBI, EAP → FW → TSBI, EAP → EI → TSBI, OAP → EU → TSBI, and OAP → FW → TSBI, thereby partially supporting H4.

## 5. Discussion

The implementation of the rural revitalization strategy and the rapid progress of the Internet have greatly boosted the development of the rural homestay inn industry. Tourists’ pursuit of rural authenticity is also partly expressed in their pursuit of authentic rural accommodation life and human interactions. The outcome of pursuit of authenticity is also partly characterized by TSBI, that is, their revisiting, recommending, or provide support, which profoundly influence the sustainable development of rural homestay inns. Previous studies have emphasized the importance of the relationships of AP with value perception [43], satisfaction [44], and place attachment [45] in understanding TSBI. However, while these scholars have also attempted to examine this issue from the perspective of tourists’ internal emotional attitudes, they failed to explore the mechanisms influencing the relationship between AP and TSBI from the interpersonal emotional perspective. Therefore, given the need for further research on this issue, this study investigated the role of the ES between rural homestay inn hosts and tourists in the way that tourists’ AP influences TSBI.

The four major findings of this study are explained in detail below.

First, AP can directly and positively influence TSBI, thereby validating H1 and the findings of previous research [43,44,45]. This result may be ascribed to the tourists’ experience of authentic local life and vernacular culture (represented by homestay inn hosts) and the genuine and natural emotions of their hosts during their interactions. These authenticity perceptions are very significant to tourists and can directly drive their TSBI toward the homestay inn, including revisiting, recommending, or providing support. In addition, OAP has a greater impact on TSBI than EAP. In other words, tourists’ perceptions of the architectural design, special meals, souvenirs, and infrastructure of the homestay inn are stronger and more likely to influence their TSBI. This finding may be due to the fact that tourists require a longer period to experience the local culture and the genuine, natural authenticity of their hosts and that they often do not spend much time in homestay inns. Therefore, the EAP is not as strong as the OAP, which can be experienced within a short period. However, this does not suggest that the EAP is less important than the OAP in influencing TSBI. 

Second, this study highlights the relationship between the effects of AP on ES. With the exception of the insignificant effect of OAP on emotional intimacy, AP has a direct positive effect on all three dimensions of ES. A high AP enhances tourists’ understanding of their homestay inn hosts, thereby strengthening their ES with these people. The absence of the direct impact of OAP on EI can be attributed to the fact that tourists do not have direct contact and do not interact with their homestay inn hosts while experiencing the objective environment of their homestay inns; thus, intimacy is not easily formed during this process [36].

Third, a significantly positive relationship was observed between tourists’ ES and TSBI. When tourists feel welcome and build emotional connections and empathy with their homestay inn hosts, they may show a stronger willingness to revisit, recommend, or support the local homestay inns. On the basis of their enhanced emotional identification, these tourists’ desire to revisit, recommend, provide support, and enjoy a similar satisfying experience may influence their future tourism decisions. This study also reaffirms the previous finding that the ES between tourists and homestay inn hosts increases the latter’s intensity of revisiting, recommending, or providing support at the destination [46].

Fourth, the mediating role of ES in the relationship between AP and TSBI was partially established. This finding extends the mechanism of the influence of AP on TSBI and innovatively examines the relationship between the two from the perspective of the host–tourist emotional relationship. Therefore, high AP can further enhance TSBI through a positive relationship with ES. The OAP can indirectly influence TSBI through EU and FW as tourists often perceive the “details” of homestay inn hosts and their unique design concepts during their experiences, which often trigger certain values that resonate with them and positively influence their TSBI. At the same time, tourists also perceive that their homestay inn hosts have prepared for their arrival with care and, as a result, these tourists feel welcome and are keen to enjoy this feeling, which would influence their future behavior. Meanwhile, the EAP can indirectly influence TSBI through EI, EU, and FW. During their stay at the homestay inn, tourists not only experience the unique culture of the countryside (including the culture that tourists identify with or pursue) but also the authentic lifestyle and pluralistic values presented by their homestay inn hosts, which, to a certain extent, trigger their empathy and identification and actively contribute to their TSBI.

## 6. Conclusions

This research empirically verified the positive influence of AP on TSBI by using rural homestay inn tourists as the research population. The three dimensions of ES were also confirmed to play mediating roles to varying degrees between the two dimensions of AP and TSBI. These findings both have theoretical and practical implications.

### 6.1. Theoretical Implications

First, this study enriches the previous research on the effects of homestay inn tourists’ AP on their TSBI and narrows the research gap from the broad homestay inn sector to a more specific one. Second, with the help of theoretical and data analyses, this study innovatively links tourists’ AP with ES, thus discovering an intrinsic link between the two and extending the possible predictor variables of ES. Third, although there is no shortage of research that predicts TSBI, there is still a paucity of research that predicts TSBI from the perspective of ES. This study sets out to fill this gap. Fourth, while previous research has attempted to explore the relationship between AP and TSBI from the perspective of tourists’ internal emotional attitudes, these works do not offer an interpersonal emotional perspective toward the mechanism between AP and TSBI. Nevertheless, the proposed research framework obtains promising empirical evidence to highlight the key role of ES in the relationship between AP and the TSBI of rural homestay inn tourists, which points to a new mechanism of influence. Although this study focuses on homestay inn tourists in a rural context, the proposed theoretical framework can be extended to rural tourism in general or other forms of tourism services and still carries general implications for research in the tourism. Future studies may test the validity of this theoretical framework in a new tourism services context.

### 6.2. Practical Implications

As a new accommodation type that differs from traditional hotels, rural homestay inns attract tourists with their objective authenticity (as reflected in their decorations that are rich in local characteristics and in the unique concept of their hosts) and existential authenticity that can meet the tourists’ authentic life experience and emotional needs, thereby promoting the sustainable development of rural tourism. The ES between hosts and tourists serves as a bridge that connects the tourists’ AP to their willingness to act, thereby clarifying the relationship between these two. The following suggestions on the design and services of rural homestay inns are thus put forward.

#### 6.2.1. Ensuring Hardware Quality and Highlighting Host Characteristics 

On the one hand, the basic function of a rural homestay inn is to serve as a place where people can rest. Therefore, a convenient and comfortable living environment is still the primary premise of homestay inn design. Ensuring good hardware facilities is conducive to reducing the negative experience of tourists and even to bring them a surprising experience. On the other hand, rural homestay inns are important carriers of rural culture. They are different from traditional hotels and are highly favored by tourists because they provide tourists the opportunity to have direct contact with the local culture and folk customs. Therefore, homestay inns should not only provide decoration, furniture, and food that can show off their local culture but also integrate the life culture of their owners as much as possible, including their hobbies, aesthetics, pursuits, and understanding of life.

#### 6.2.2. Enhancing the Interaction between Hosts and Tourists and Creating an Emotional Connection

Homestay inn owners should identify and meet the diverse needs of tourists and provide thoughtful, warm, and even unexpected services in order for the latter to feel the enthusiasm and sincerity of the former. These hosts should also take the initiative to create conditions and opportunities, such as holding a sharing meeting exclusive to the homestay inn, encouraging tourists to participate in local homestay inn activities, actively communicating and interacting with tourists, and sharing their attitudes toward life and the cultural concepts to be conveyed by their homestay inns. By sharing their sincere feelings to tourists, tourists will not only gradually establish an emotional connection with their hosts but also get to know them better in their process of communication, thereby strengthening their sense of identity and stimulating their willingness to revisit or recommend the homestay inn to others.

#### 6.2.3. Inheriting the Concept of Homestay Inns and Ensuring the Management Level

During their development process, many rural homestay inns tend to fall into a trap, that is, they cannot maintain their authenticity after their expansion. This phenomenon lies in the absence of the previous host, who relegates the task of defining the design and service of these inns to others, thereby depriving these establishments of an “internal cultural soul,” making their services too standard and single, and eventually turning these inns into simple accommodations that are no different from traditional hotels. Therefore, during the development and expansion of rural homestay inns, owners should pay attention to their construction of a talented team, ensure that the hosts have certain cultural cultivation and their own pursuits, maintain their love for their establishment, acknowledge their culture, and willingly interact with tourists to convey such culture. Only by ensuring the quality of homestay inn operators can people inherit the homestay inn concept to the greatest extent and improve the accommodation experience of tourists.

## 7. Limitations and Future Research

This study follows rigorous academic norms and relevant data analysis operations, but some shortcomings still need to be noted. First, in terms of analysis methods, although this study has improved the established scale in the preliminary analysis, the design of this scale is still based on previous scales with minor modifications, and some slight shortcomings may be observed in the accuracy and completeness of the study in rural homestay inn scenarios. Therefore, future studies may consider conducting qualitative interviews to extract highly scientific and targeted measurement indicators and improve the relevant research. Second, this study was not conducted in a very timely manner. Moreover, this study did not discuss the different types of rural homestay inn tourists and the differences in the degree of ES between tourists of different genders, ages, personalities, and nationalities, which could also serve as an effective tool for further improving the relevant service content. Future research may therefore explore this area further. Third, given that the case sites in this study are only rural homestay inns and do not involve other types of homestay inns, such as urban homestay inns, the findings may not be directly generalizable to the general homestay inn context. Further research is needed to verify the broad applicability of the proposed theoretical framework.

## Figures and Tables

**Figure 1 behavsci-12-00341-f001:**
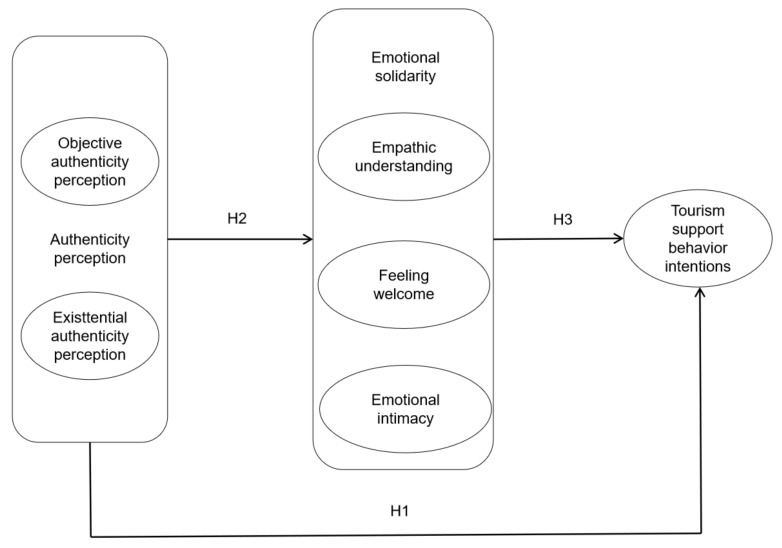
Research model.

**Figure 2 behavsci-12-00341-f002:**
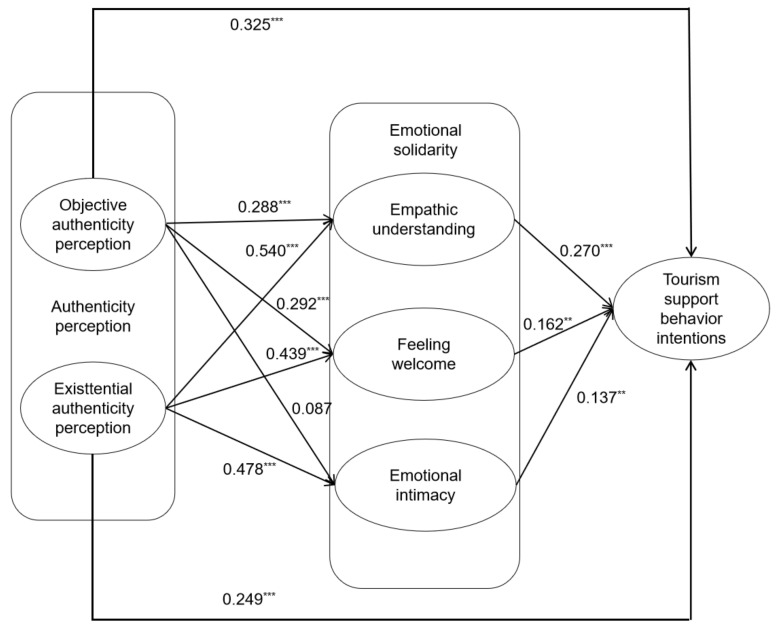
Results of the hypotheses test. ** *p* < 0.01, *** *p* < 0.001.

**Table 1 behavsci-12-00341-t001:** Exploratory factor analysis.

Variable	Items	Description	Factor Loading	Eigenvalues	Cumulative Variance Explained	CITC	Cronbach’s α
EAP	EAP1	The homestay inn has a warm, cozy, home-like atmosphere	0.835	4.071	37.008	0.757	0.904
	EAP2	This stay gave me a better understanding and appreciation of the local life and culture	0.764			0.705	
	EAP3	I got to experience the authentic life of a local homestay inn host	0.869			0.809	
	EAP4	The host of this homestay inn is hospitable and attentive	0.816			0.728	
	EAP5	I can feel the authentic and sincere natural emotions of the hosts as I interact with them	0.764			0.700	
	EAP6	Coming to this homestay inn has driven me to think about my lifestyle and attitude toward life	0.814			0.720	
OAP	OAP1	I like the architecture and design of the homestay inn	0.720	3.230	66.373	0.539	0.838
	OAP2	This homestay inn is popular for its locally inspired cuisine	0.798			0.672	
	OAP3	I like that the homestay inn offers local souvenirs and other souvenirs for purchase	0.832			0.766	
	OAP4	The homestay inn has a clean and well-equipped infrastructure	0.762			0.568	
	OAP5	I feel that the overall architectural style of the homestay inn is in harmony with the surroundings and the building complex	0.798			0.678	
KMO = 0.854, Bartlett (df = 55, *p* = 0.000)							
EU	EU1	I think that the hosts of the homestay inn would appreciate the financial contribution that my stay would bring to the homestay inn	0.852	3.087	30.869	0.794	0.910
	EU2	I think that the hosts of the homestay inn will appreciate all the benefits that my stay will bring to the local area	0.898			0.883	
	EU3	I felt that the host of the homestay inn was very kind to me	0.784			0.743	
	EU4	I made friends with the host of the homestay inn	0.807			0.766	
FW	FW1	I think that the hosts of the homestay inn would appreciate the financial contribution that my stay would bring to the homestay inn	0.816	2.910	59.970	0.755	0.882
	FW2	I think that the hosts of the homestay inn will appreciate all the benefits that my stay will bring to the local area	0.854			0.781	
	FW3	I felt that the host of the homestay inn was very kind to me	0.821			0.684	
	FW4	I made friends with the host of the homestay inn	0.767			0.770	
EI	EI1	I think that the hosts of the homestay inn would appreciate the financial contribution that my stay would bring to the homestay inn	0.903	1.802	77.989	0.759	0.863
	EI2	I think that the hosts of the homestay inn will appreciate all the benefits that my stay will bring to the local area	0.879			0.759	
KMO = 0.843, Bartlett (df = 45, *p* = 0.000)							
TSBI	TSBI1	I would choose this homestay inn again for my next trip if I had the chance	0.879	3.042	76.043	0.781	0.893
	TSBI2	I would like to speak positively about and promote this homestay inn to others	0.855			0.752	
	TSBI3	I would like to share my pleasant stay on online social media platforms	0.876			0.737	
	TSBI4	If someone asks for advice, I would recommend this homestay inn to them	0.878			0.791	
KMO = 0.807, Bartlett (df = 6, *p* = 0.000)							

**Table 2 behavsci-12-00341-t002:** Confirmatory factor analysis.

Variable	Items	Factor Loading	AVE	CR
		>0.5	>0.5	>0.7
EAP	EA1	0.809	0.620	0.907
	EA2	0.758		
	EA3	0.867		
	EA4	0.793		
	EA5	0.727		
	EA6	0.762		
OAP	OA1	0.647	0.550	0.858
	OA2	0.714		
	OA3	0.851		
	OA4	0.766		
	OA5	0.713		
EU	SU1	0.824	0.705	0.905
	SU2	0.940		
	SU3	0.751		
	SU4	0.832		
FW	FW1	0.775	0.646	0.879
	FW2	0.822		
	FW3	0.765		
	FW4	0.850		
EI	EC1	0.810	0.753	0.859
	EC2	0.922		
TSBI	SB1	0.839	0.680	0.895
	SB2	0.796		
	SB3	0.806		
	SB4	0.856		

**Table 3 behavsci-12-00341-t003:** Tests of discriminant validity.

Variable	1	2	3	4	5	6
EAP	0.787					
OAP	0.355 **	0.742				
EU	0.482 **	0.370 **	0.840			
FW	0.452 **	0.301 **	0.505 **	0.804		
EI	0.431 **	0.243 **	0.435 **	0.389 **	0.868	
TSBI	0.564 **	0.486 **	0.652 **	0.515 **	0.479 **	0.825

Note: The values on the diagonal line are the square roots of AVE, whereas those off the diagonal line are the inter-construct correlation coefficients. ** *p* < 0.01.

**Table 4 behavsci-12-00341-t004:** Path analysis.

Path	Estimate	S.E.	t
EAP → EU	0.540 ***	0.080	6.736
EAP → FW	0.439 ***	0.066	6.673
EAP → EI	0.478 ***	0.070	6.786
OAP → EU	0.288 ***	0.078	3.675
OAP → FW	0.292 ***	0.071	4.137
OAP → EI	0.087	0.077	1.121
EAP → TSBI	0.249 ***	0.066	3.755
OAP → TSBI	0.325 ***	0.060	5.387
EU → TSBI	0.270 ***	0.047	5.695
FW → TSBI	0.162 **	0.057	2.844
EI → TSBI	0.137 **	0.048	2.837

** *p* < 0.01, *** *p* < 0.001.

**Table 5 behavsci-12-00341-t005:** Mediating effect analysis.

Path	Estimate	S.E.	95% CI
EAP → EU → TSBI	0.245	0.037	(0.173,0.317)
EAP → FW → TSBI	0.188	0.041	(0.113,0.272)
EAP → EI → TSBI	0.128	0.030	(0.072,0.192)
OAP → EU → TSBI	0.247	0.044	(0.161,0.334)
OAP → FW → TSBI	0.202	0.045	(0.115,0.293)

## Data Availability

The data used to support the findings of this study are available from the corresponding author upon request.

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
