# Peer review of "A Study on the Mediating Role of Emotional Solidarity between Authenticity Perception Mechanism and Tourism Support Behavior Intentions within Rural Homestay Inn Tourism"

_behavsci, 2022, doi:10.3390/bs12090341_

Round 1

Reviewer 1 Report

The work is interesting and with contributions to academia and organizations operating in the sector.

I believe the references should be updated, with more references from 2021 and 2022.

On the other hand, the limitations identified by the authors refer only to the methodology used, especially the scale used, recommending, in the future, conducting interviews. Authors should be more explicit about limitations and recommendations.

Good luck

Author Response

Dear Editor and Dear reviewer,

Thanks very much for taking your time to review this manuscript. We really appreciate all your generous comments and suggestions! The following is our response to all your suggestions. Please check:

According to your suggestions, we updated several references in 2021. At the same time, we further explained the limitations of the article in the article's limitations and future research, and gave relevant suggestions.we are more explicit about limitations and

Thank you for your careful review. We really appreciate your efforts in reviewing our manuscript during this unprecedented and challenging time. We wish good health to you, your family, and community. Your careful review has helped to make our study clearer and more comprehensive.

Author Response

Dear Editor and Dear reviewer,

Thanks very much for taking your time to review this manuscript. We really appreciate all your generous comments and suggestions! All of your questions were answered one by one.

  1. According to the suggestions, we have revised the contents of the abstract to include clear research objective.
  2. Where the TRA theory first appeared, it has been fully spelled, and readers can understand it more quickly.
  3. Relevant suggestions have been added into the article.
  4. Inappropriate fonts have been modified.
  5. In the discussion part, we have made more expansion to make the content of the article more detailed.
  6. The replicability of the research has been mentioned in the theoretical enlightenment.
  7. We have changed “thisstudy”to “this study”.
  8. We have shown the analysis results in the form of pictures in Figure 2.

Thank you for your careful review. We really appreciate your efforts in reviewing our manuscript during this unprecedented and challenging time. We wish good health to you, your family, and community. Your careful review has helped to make our study clearer and more comprehensive.

Reviewer 3 Report

To Author(s)

There are many words that are common substantives and should not be capitalized after a comma. The reverse applies to the full stop, where all words after the full stop must begin with a capital letter. Please check the entire text and correct accordingly.

There are several other parts that I suggest should be fixed. Are they:

Line 11: In the first line of the Abstract there is redundancy in the terms used. In the following extract this is markedly visible “…tourism, and tourists’ tourism…”.

L 15: “TRA theory” – As this is the first time this abbreviation appears, and as not all readers are familiar with the term, it is important to spell out the words contained in the abbreviation.

Ls 16-23: Although the Abstract is presented in only one paragraph, the sentence that appears between lines 16 to 23 is too long. A sentence to be understood must have a maximum of 6 lines. Please divide the sentence into 2 or 3.

The same type of problem appears more often. It is advisable to split the text into several sentences. Here are some examples where this happens: Ls: 208-215, 324-330, maybe others.

Ls 60-65: This part of the text should be allocated elsewhere, but not in the “Introduction”. May be in the M&M section.

L 61: Please correct the typo “inntourists”. Note: there are several other typos like this that should be amended.

L 68: The word “Sexuality” is a word that is out of context. This is a typo that should be amended.

Ls 76-78: Identically, this part of the text should be allocated elsewhere, but not in the “Introduction”. May be M&M.

Ls 105-109: Please check this sentence. It seems that some bits should be amended. Otherwise, it seems the sentence does not make sense.

The “Methodological” part is quite poor. The part referring to the questionnaire is scarce. Age groups are not very convincing. These parts jeopardise following statistical analyses.

Section 4.1 is not well formatted…

L 255-256: Men and women is not exactly proportional. The emphasis given is slightly incorrect.

L 270: There is a typo that should be amended.

Table 1: “I became friends” is not correct. Please amended it accordingly. Note: It appears more than once.

Throughout the text the words “behaviour”, “behavior”, “behavioural”, “behavioral” appear in all these forms. They should appear in just 2 forms.

L 316: “Secondary” should be substituted by “Secondly”.

Author Response

Dear Editor and Dear reviewer,

Thanks very much for taking your time to review this manuscript. We really appreciate all your generous comments and suggestions! The following is our response to all your suggestions. Please check:

  1. We have carefully corrected the problems such as redundant terms, long sentences, misspellings, and language defects in the articles you mentioned.
  2. As for the tra theory that first appeared in the article, all the words have been spelled out, which is convenient for readers to understand.
  3. Some sentences have been assigned from the introduction to other more appropriate parts, strengthening the overall logic of the article.
  4. As for the " ‘methodological’part is quiet poor. The part referring to the questionnaire is scar" mentioned by you, during the process of writing such contents as using crawler technology, expert evaluation and Advance research deletion, we found that there are too many contents in this part. In consideration of the convenience of readers' reading, it is not appropriate to display a large part on the content of the article, so we simplified it as much as possible, but basically listed the due procedures, The deleted items and the final questionnaire are also presented below.

Thank you for your careful review. We really appreciate your efforts in reviewing our manuscript during this unprecedented and challenging time. We wish good health to you, your family, and community. Your careful review has helped to make our study clearer and more comprehensive.

Reviewer 4 Report

COMMENTS - REVIEWER 1

ARTICLE : [Behavioral Sciences] Manuscript ID: behavsci-1886352

Title: A Study on the Influence Mechanism of Authenticity Perception and Tourism Support Behavioral Intentions of Rural Homestay Inn Tourists - Mediated by Emotional Solidarity

        First of all, I would like to point out that, in my opinion, the present study is a fairly good one in terms of quality and that its content complies with most of the standard requirements/conditions that exist for the design and production of such scientific articles. In essence, the authors propose their own model, (figure 1, page 5) which is based on the framework provided by TRA (Theory of Reasoned Action) to analyse TSBI (Tourism Support Behavioral Intentions) in rural tourism in China, Fujian province, with reference to Rural Homestay Inns. At the basis of the study is a significant amount of work done by the author to design a questionnaire, apply it (in Fujian province, in a historical city where tourist establishments offering inn type accommodation predominate and there is a local/cultural context that highlights some of the authenticity characteristics of Chinese culture), obtain 257 valid results, interpret them, etc. It is important to highlight, I think, the field research carried out by the authors of the study (with TRA as a starting point). The present study is partly based on descriptive statistics (tables 1, 2, 3, 4, 5, page 7-10) and follows the typical structure, in the sense that 4 hypotheses are stated, the necessary statistical tests are applied, the hypotheses/results are interpreted, etc. If the proposed study is accepted for publication, it would be of interest to specialists in different fields (psychology, sociology, organisational behaviour, marketing, tourism, etc.) and even to the general public.

         In my opinion, the main limitation of the present study is the poor/modest theoretical grounding at the beginning of the study and the quality of the bibliographical references at the end of the study (out of the total 42 references, only 3 are books/volumes and the others are only small articles/studies). In order to improve the quality of the proposed study, I suggest some further issues to the authors:

A. Content aspects

To improve the content of the study I suggest:

1.         The authors have constructed quite a good argument in connection with the application of TRA in the relationship between tourists and hosts in rural tourism, with reference to the accommodation in inns prevailing in a historical city in Fujian province. Indeed, I believe that by using TRA (and possibly other theories proposed by sociology/psychology) a climate based on mutual trust and empathy between tourists and hosts can be built, which could lead to Emotional Solidarity as a mediating factor between Authenticity Perception Mechanism (APM) and Tourism Support Behavioral Intentions (TSBI). However, the authors hardly explain the content of the TRA, even while using this theory as a framework for the whole study. They make a single reference to authors Fishbein Martin and Ajzen Icek (reference 18, at the end) even although these two have several published volumes and are unanimously considered to be the founders of TRA.

      (I am not referring to the fact that the bibliographical reference in item 18 is inaccurate as the authors have taken two words from the title of the article and unintentionally created a third author, which does not actually exist; the study mentioned is a significant one and published in an important journal, but insufficient to reflect the much broader conception of the two authors). Also in the sense claimed, I point out that the authors refer extensively to the concept of EI (Emotional Intelligence) from page 1, point 1; page 3 point 2.2 ; various components of the model in figure 1 etc.; they do not, however, explain in advance the basic idea of EI and its relation to TRA.

In my opinion, in point 1, page 2, line 66, after the phrase "Academics...", the authors should add at least 15-16 additional sentences to illustrate:

a.         In the structure of 10-12 sentences, the history and content of the TRA should be briefly described, particularly the contribution made by Fishbein Martin and Ajzen Icek, from the 1960s to the present; at least two works should be briefly discussed:

- Fishbein M. and Ajzen I., Beliefs, attitude, intention and behavior: An Introduction to theory and research, Addison-Wesley Publishing Company, 1975 ( about 580 pages text would obviously provide a more complete documentation than the study with the same title , about 20 pages , indicated by the authors in reference 18 , final );

- Ajzen I., Attitudes, Personality and Behavior, McGraw-Hill Education, 2005.

b.         In the structure of another 4-5 sentences the concept of EI should be briefly outlined, in particular the role of empathy in social psychology, the relationship between the individual and the group, etc.; this concept is well known to the general public; see, for example, D. Goleman, Emotional Intelligence, Bantam Books, 1995.

Argument: Assuming that the authors extend the theoretical justification for TRA , on which the whole study is based, this should bring more clarity and credibility to their proposed model (Figure 5), the reasoning they constructed and the results they arrived at. In addition, the succinct definition of EI would, I believe, ensure a greater attractiveness of the study in the hypothesis of its publication (since this concept has become a benchmark in modern society). However, the authors refer extensively to the importance of empathy to achieve Emotional Solidarity in the relationship between tourists and hosts in the mentioned province and with reference to accommodation in inns. In addition, the substantiated presentation of TRA would also be of interest to specialists in other fields such as consumer behaviour, organisational behaviour, marketing, organisational culture, etc.

2 The interpretation of the questionnaire data, the application of statistical tests, starting with Table 1, page 7, are correct in content; I suggest that in Table 1 each variable, with the 4-5 items, be more clearly delineated from each other (as the authors did in Table 2). I also suggest the authors, after Table 4 and Table 5, to add 1-2 additional sentences explaining more clearly for the reader the relevance of the data in the tables.

3    The structure of the whole study needs to be reviewed and correctly annotated as from point 4, pages 6-10, the following is also annotated point 4, the discussion part

      4.   In relation to the wording of the hypotheses underlying the study, i.e. 2.1. and 2.2., I suggest the authors to replace the phrase "Based on this..." on page 4 when stating H2 - H4  with an expression such as: "In addition to hypothesis H1 mentioned above, we further state 3 other hypotheses underlying the study" (to avoid redundancy of the text in the current wording).

5.         On page 6, point 4.1., starting with line 262, the income groups for respondents should be indicated in USD or EURO as the survey is submitted for publication in English and to an international journal that is aimed at readers familiar with USD.

6.         I have carefully reviewed the study received for evaluation and believe it is good in content. At the same time, I think the title of the study is slightly confusing/unclear; I suggest the authors to consider the hypothesis of rephrasing the title to be more attractive to potential readers; for example, a hypothetical variant could be: "A study on the Mediating role of Emotional Solidarity between Authenticity Perception Mechanism and Tourism Support Behavioral Intentions within Rural Homestay Inn Tourism"

B. Formal aspects

With reference to some formal errors/corrections I point out:

1.         Authors should carefully check the entire text for misspellings/grammatical corrections (e.g.: page 1, line 34, the sentence "Tourists expect...", starts with a small letter; page 3, line 100, before "At the same time..." is probably a full stop; page 5, line 211, the phrase "Fujian on Ctrip..." does not make sense, etc.).

2.         On page 2, line 67-69, the sentence beginning "Emotional attitudes have..." does not make sense; it should be rephrased completely.

3.         In my opinion, some sentences are too long and induce confusion for the reader (even if the idea sustained by the authors is inferred); for example:

- in the abstract, the sentence beginning "The results of..." should be rephrased and split into 3-4 separate sentences;

- on page 12, the sentence beginning "At the same time..." should be split into 2-3 separate sentences.

Author Response

Dear Editor and Dear reviewer,

Thanks very much for taking your time to review this manuscript. We really appreciate all your generous comments and suggestions! The following is our response to all your suggestions. Please check:

  1. In content aspects, you pointed out that the TRA theory in our article was too brief, and we made some low-level mistakes in literature reference. We have made amendments to all of these. First of all, we have adopted your suggestions on references, replaced simple journal papers with corresponding books, and confirmed the author's name again.  Secondly, we supplemented the relevant contents of TRA appropriately, but did not add too many contents. The reason is that the research hypotheses of this paper are mainly based on the inheritance of previous scholars' research hypotheses, and the relationships between variables are respectively demonstrated, which is the main basis for establishing the theoretical framework of this paper. The reason why this paper introduces TRA theory into this paper is that as a mature theory, it can further corroborate the theoretical framework proposed in this study. However, because it is not the core content of this paper, it does not spend a large amount of time to describe this theory. Of course, this is only our idea, and there may be some deficiencies in the writing of the article. Therefore, we adopted your suggestion, expanded the contents of TRA theory, and reorganized the sentences. In addition, for the interpretation of the questionnaire data you mentioned, we have also adopted your suggestions and made supplementary explanations in the corresponding paragraphs of Table 4 and table 5, so that readers can understand the meaning more easily. Finally, we have revised the 3-6 mentioned by you.
  2. In formal aspects, we have carefully revised some meaningless words and sentences and some long sentences to make their expression more accurate and concise.

Thank you for your careful review. We really appreciate your efforts in reviewing our manuscript during this unprecedented and challenging time. We wish good health to you, your family, and community. Your careful review has helped to make our study clearer and more comprehensive.

Round 2

Reviewer 3 Report

Although several inaccuracies have been corrected in relation to the previous version of the manuscript (MS), there are still many parts that need to be revised and edited in order to have greater coherence in the text. There are still incomplete words lost in the text…

In the previous version - i.e., before the 1st revision - there were two sections 4 by lapse. After the revision this lapse disappeared, but the Discussion was now collated to the Conclusion. In my opinion this should not have been done, as the Discussion and Conclusion should be kept as separate sections. Only the section number should have been amended.

I believe that the review needs to be done more carefully. It doesn't seem to me that doing a very fast review is synonymous of a good review. On the contrary, the review seemed to me to be done too hastily and carelessly.

I suggest that, in order to be publishable, authors should pay more attention to editing the text and eventually ask for help from someone who is native or at least familiar to the English language.

Author Response

Dear Editor and Dear reviewer,

Thanks very much for taking your time to review this manuscript again. We really appreciate all your generous comments and suggestions! All of your questions were answered one by one.

  1. The Discussion and Conclusionshave been divided into two parts according to your suggestions, and the section numbers have been added respectively.
  2. We are very lucky to receive your honest and fair criticism. In the first review, we were a little hasty and careless. After accepting your suggestions, we asked our professor for help, and then we sent the manuscript to an English speaking long-term partner. He spent several days reviewing and editing the manuscript in detail, and then we received the manuscript after polishing. We further checked and revised the polished manuscript, and finally got the final manuscript.

Thank you for your careful review again. We really appreciate your efforts in reviewing our manuscript during this unprecedented and challenging time. We wish good health to you, your family, and community. Your careful review has helped to make our study clearer and more comprehensive.

Round 3

Reviewer 3 Report

To Author(s)

The manuscript (MS) was duly proofread by someone native (or close)  to the English language, which evidently greatly improved its reading.

As the MS took a big change in terms of content, I suggest that now in the final edition only a few minor inaccuracies or duplicate words – which are still sometimes remaining – in the same sentence should be revised.

Author Response

Dear reviewer:

Thank you for reviewing our manuscript, titled “A study on the mediating role of emotional solidarity between authenticity perception mechanism and tourism support behavior intentions within rural homestay inn tourism” and providing us with the comments and suggestions. We have tried our best to revise the paper accordingly by addressing your comments (the changes are highlighted in the revised paper). In order to facilitate reading, I uploaded two manuscripts, one with track changes and the other without track changes.

We wish the revised paper would fulfill the standard of the Behavioral Sciences for publication.

Yours sincerely,

The Authors

Response:

  1. We agree with you that there are still a few minor inaccuracies or duplicate words in the manuscript. Therefore, we invited another English expert to check the grammar and use of words in the manuscript in detail and made further revisions.

For example,

in line 19, change "these prior studies" to "they".

In line 22, deleted "and" before “constructs”.

In line 41,44,54 In order to avoid the same word appearing too many times, we replace "authentic" with "genuine", "distinctive”, “unpretentious” and “spontaneous”.

In line 58, change "individuals" to "people".

In line 91, change "and" to "which".

In line 130, delete the previous "individual".

In line 133, change "emotion" to "feeling".

In line 158, delete the second "in".

In line 160, change "feeling close" to "closeness to tourists".

In line 332, delete the "authentic".

In line 371, change "emotionally" to "satisfying".

In line 387, delete the "and unpretentious".

...

Such changes have been highlighted in the revised manuscript.

  1. In order to make the sentences in the manuscript shorter and easier for readers to understand, we have replaced some concepts that appear repeatedly in the manuscript with abbreviations, such as "authenticity perception" abbreviated as "AP", "emotional solidarity" abbreviated as "ES", and "tourism support behavioral ideas" abbreviated as "TSBI", etc.